# The genomic landscape of pediatric myelodysplastic syndromes

Jason R. Schwartz[1], Jing Ma[2], Tamara Lamprecht[2], Michael Walsh[2], Shuoguo Wang[3], Victoria Bryant[2], Guangchun Song[2], Gang Wu [3], John Easton[3], Chimene Kesserwan[1], Kim E. Nichols[1], Charles G. Mullighan [2], Raul C. Ribeiro[1] & Jeffery M. Klco [2]

Myelodysplastic syndromes (MDS) are uncommon in children and have a poor prognosis. In contrast to adult MDS, little is known about the genomic landscape of pediatric MDS. Here, we describe the somatic and germline changes of pediatric MDS using whole exome sequencing, targeted amplicon sequencing, and/or RNA-sequencing of 46 pediatric primary MDS patients. Our data show that, in contrast to adult MDS, Ras/MAPK pathway mutations are common in pediatric MDS (45% of primary cohort), while mutations in RNA splicing genes are rare (2% of primary cohort). Surprisingly, germline variants in *SAMD9* or *SAMD9L* were present in 17% of primary MDS patients, and these variants were routinely lost in the tumor cells by chromosomal deletions (e.g., monosomy 7) or copy number neutral loss of heterozygosity (CN-LOH). Our data confirm that adult and pediatric MDS are separate diseases with disparate mechanisms, and that *SAMD9/SAMD9L* mutations represent a new class of MDS predisposition.

[1] Department of Oncology, St. Jude Children's Research Hospital, 262 Danny Thomas Place, Mail Stop 260, Memphis, TN 38105, USA. [2] Department of Pathology, St. Jude Children's Research Hospital, 262 Danny Thomas Place, Mail Stop 342, Memphis, TN 38105, USA. [3] Department of Computational Biology, St. Jude Children's Research Hospital, 262 Danny Thomas Place, Mail Stop 1135, Memphis, TN 38105, USA. Jason R. Schwartz and Jing Ma contributed equally to this work. Correspondence and requests for materials should be addressed to J.M.K. (email: jeffery.klco@stjude.org)

Myelodysplastic syndromes account for <5% of pediatric hematologic malignancies with an incidence of 2–4 cases/million[1]. The prognosis of pediatric MDS is typically poor, because cytotoxic chemotherapies like those used to treat acute leukemia are not successful, thus leaving only bone marrow transplantation as a curative option[1,2]. Much has been learned about adult MDS through next-generation DNA sequencing. Multiple large cohort studies of adult MDS patients found recurrent mutations in genes important in epigenetic regulation (e.g., TET2, ASXL1, and DNMT3A), and RNA splicing (e.g., SF3B1 and U2AF1)[3–7]. However, in limited studies on pediatric MDS, mutations in these genes are uncommon[8,9]. This is not surprising as there are well-accepted clinical and morphologic differences between pediatric and adult MDS (e.g., bone marrow hypocellularity is more common in children) leading the World Health Organization (WHO) to classify MDS differently in adults and in children[10].

It is becoming increasingly recognized that germline variants in different transcription factors, such as GATA2, RUNX1, ETV6, or CEBPA, can lead to familial MDS/AML[11–14]. In particular, germline GATA2 variants have been shown to occur in 7% of pediatric primary MDS[15]. In contrast, other studies using targeted sequencing of children with idiopathic bone marrow failure or MDS found pathogenic variants in only approximately 10% of patients[16], suggesting the need for more comprehensive sequencing. The spectrum of genes harboring germline variants in pediatric MDS has also recently begun to expand beyond transcription factors, including ANKRD26[17] and SRP72[18]. Recently, germline variants in SAMD9 and SAMD9L have been reported in clinical syndromes affecting multiple organ systems that are also associated with MDS and monosomy 7[19-22], as well as isolated familial MDS[23].

Despite this progress, no study to date has performed comprehensive sequencing on a pediatric MDS cohort to fully understand somatic and germline variation in this neoplasm. In this study we perform tumor and normal whole exome sequencing (WES) on 32 pediatric primary MDS patients and targeted sequencing on another 14 cases through a single institution study focused on defining the genomic landscape of pediatric MDS. For comparison, we similarly characterize 23 cases with overlapping features of MDS and myeloproliferative neoplasm (MDS/MPN), namely juvenile myelomonocytic leukemia (JMML), and 8 cases of AML with myelodysplasia-related changes (AML-MRC). We show that Ras/MAPK pathway mutations are common in pediatric primary MDS (45%) while mutations in RNA splicing genes are rare (2%), and that germline SAMD9/SAMD9L mutations are present in 17% of primary MDS patients. These data suggest that pediatric MDS is separate from adult MDS with disparate underlying mechanisms.

## Results

**Sequencing of pediatric MDS samples.** We performed next generation sequencing on a cohort of 77 pediatric patients with diagnoses of primary MDS (n = 46), MDS/MPN (n = 23, 19 of which were JMML), and AML-MRC (n = 8) (Fig. 1, Supplementary Data #1 and Supplementary Fig. 1). Patients with a confirmed bone marrow failure syndrome, Down syndrome, or therapy-related MDS were excluded from this cohort. Three siblings with MDS that we recently described were included[23]. Of the primary MDS patients, 50% were classified as refractory cytopenia of childhood (RCC) and 50% as refractory anemia with excess blasts (RAEB). Paired tumor-normal WES was performed on 54 of these cases with a median coverage of ×96 in the normal and ×92 in the tumor (Table 1, Supplementary Datas #2 and 3). A subset of the variants was then validated by amplicon

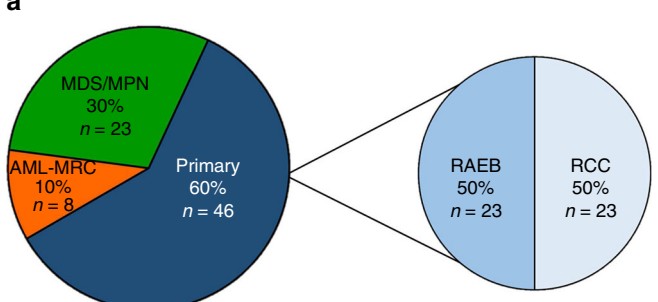

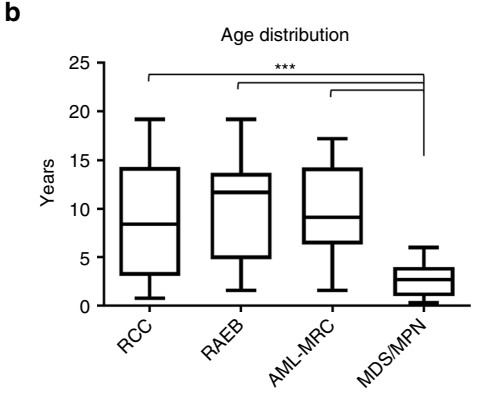

**Fig. 1** Pediatric MDS cohort. **a** Pie charts depicting the distribution of the three diagnostic categories and subcategories of the pediatric MDS cohort; Total: n = 77. **b** Distribution of age (in years) at diagnosis for the pediatric MDS cohort. Lines within boxes represent median age (RCC: 8.4, RAEB: 11.7, AML-MRC: 9.1, MDS/MPN: 2.7) and whiskers represent maximum and minimum. ***: p < 0.0001 (student's t-test). AML-MRC AML with myelodysplasia-related changes, RAEB, Refractory anemia with excess blasts; RCC, Refractory cytopenia of childhood

sequencing to a mean depth of ×7000. CD3 + or unfractionated lymphocytes were flow sorted from the diagnostic sample to use as the source of normal comparator genomic DNA (gDNA) as other sources of normal gDNA were not available (Supplementary Fig. 2). This strategy has been successfully used in the past for JMML[24] and other myeloid neoplasms[25]. Tumor-only material was available for the remainder of the cohort (n = 23), which was sequenced using a custom amplicon strategy targeting recurrent mutations identified in the WES cohort and other hotspots from adult and pediatric myeloid neoplasms with a median coverage of ×3000 (Supplementary Datas #4–6). In addition, RNA-sequencing was performed on 43 cases with available high-quality RNA (mean mapped reads: 107 million). For the 54 cases with WES, the somatic mutation rate and allele burden varied between the disease types (Fig. 2a, Supplementary Fig. 3) with the lowest number of mutations observed in RCC (mean: 4; range: 0–11). Out of the 18 RCC cases and 14 RAEB cases sequenced by WES, 3 RCC cases and 1 RAEB case demonstrated no somatic coding mutations, 2 of which contained potential causative germline variants (see below). Furthermore, our data show that mutations in genes important in the Ras/Mitogen-activated protein kinase (MAPK) pathway are the most common in our cohort while mutations in genes involved in RNA splicing are rare (Fig. 2b). Several other patients have no somatic mutations in genes typically implicated in myeloid neoplasms (Fig. 2c). The mutational signature for these cases was similar to that previously observed for AML (Signature 1A/1B), characterized by prominence of C > T substitutions at NpCpG trinucleotides (Supplementary Fig. 4)[26].

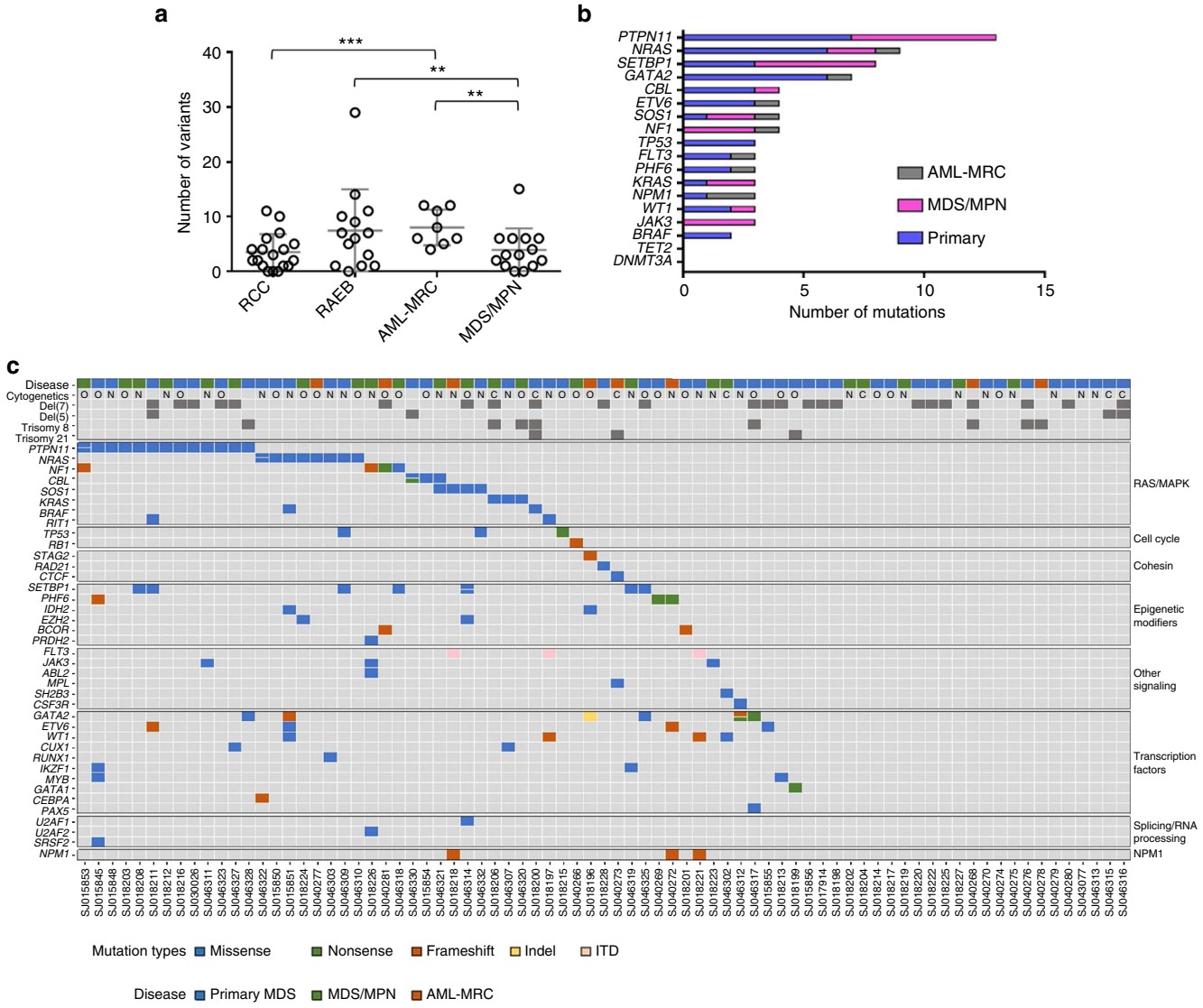

**Fig. 2** Somatic mutations in pediatric MDS and related neoplasms. **a** Total number of somatic variants per patient in the 54 patients with WES data (includes silent, nonsense, missense, frame shifts, indels, ITD, and mutations within 3′ and 5′ UTR). **: $p = 0.02$; ***: $p = 0.003$ (student's t-test). **b** The most common genes with somatic mutations in the full cohort of 77 patients (includes WES and targeted amplicon sequencing). Only somatic mutations with presumed functional consequences are shown. **c** Heat map showing the somatic mutational profile of the pediatric MDS cohort separated by gene functional groups. Only somatic mutations with presumed functional consequences are shown. Split cells indicate multiple mutations. O, other karyotype findings not listed separately; C, complex karyotype; N, normal karyotype

**Copy number alterations in pediatric MDS**. Copy number information, obtained from WES data and conventional karyotyping, determined that deletions involving chromosome 7 were more frequent in primary MDS ($n = 19$, 41%) than in MDS/MPN ($n = 3$, 13%) and AML-MRC ($n = 2$, 25%) (Fig. 3). Approximately 60% of RCC cases had deletions involving chromosome 7 (13 of 23), compared to only 26% of RAEB cases (6 of 23). Deletions involving chromosome 5 were infrequent ($n = 4$, 5%); all occurrences were in RAEB cases. Trisomy of chromosomes 8 ($n = 8$, 10%) and 21 ($n = 3$, 4%), and deletions or loss of heterozygosity of 17 ($n = 2$, 3%) were present at low frequency within the total cohort. In total, we detected 17 additional copy number abnormalities (including 3 cryptic chromosome 7 abnormalities) with WES that were not reported by standard conventional karyotyping with an average size of 40 Mbp (Range: 0.02–159 Mbp) (Supplementary Data #7). Alternatively, subclonal copy number alterations identified by conventional karyotyping ($n = 12$) were not found with WES. Notably, in 2 cases, subclonal

del(7) was identified in 3 of 20 metaphases analyzed, yet was undetectable in the WES data. Copy number neutral events were identified in the lymphocytes (source of germline) in 3 patients, with involvement of chromosome 7 (x2) or 17 (Supplementary Fig. 5). The 7q CN-LOH event in SJ18228 has been previously reported by our group[23].

**Putative germline variants in pediatric MDS**. We analyzed the 54 tumor-normal pairs for the presence of germline variants to determine if any cases harbored pathogenic variants that may predispose patients to a myeloid malignancy. It is important to note that our normal sample is from flow cytometry purified lymphocytes and thus cannot be used to definitively categorize a variant as germline given that it is possible for somatic mutations to arise in progenitor compartments affecting both myeloid and lymphoid cells, as has been shown in JMML[27,28]. We first analyzed all coding variants in over 1000 genes that have been implicated in pediatric cancer predisposition[29], JMML[30], familial

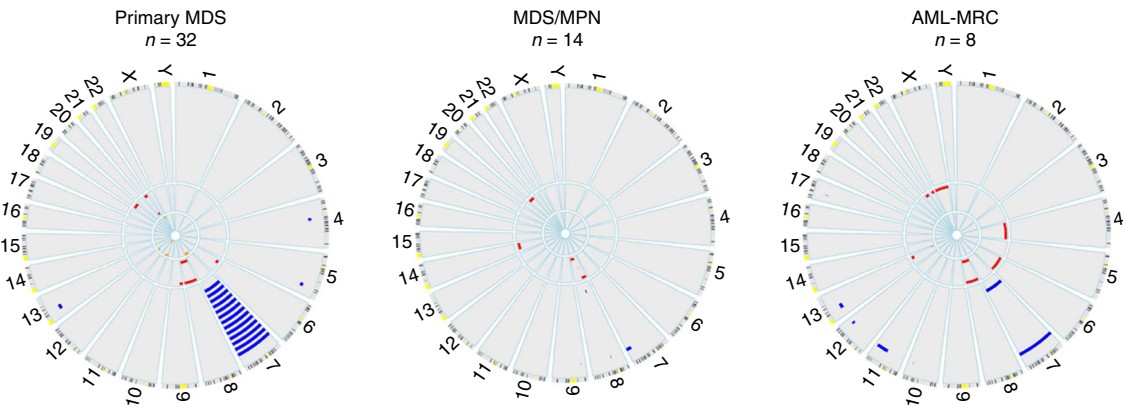

**Fig. 3** Copy number analysis. Circos plots showing copy number alterations found with WES analysis of 54 tumor-normal pairs (events identified by conventional karyotyping are not included in these plots). Circumferential numbers indicate chromosome number, blue lines (outside ring) = deletions, red lines (middle ring) = amplifications, and orange lines (inside ring) = copy number neutral-loss of heterozygosity

**Table 1 Characteristics of the Pediatric MDS Cohort**

|  | Total | DNA Seq. Technique | RNA Seq. |
|---|---|---|---|
| *Tumor/Normal Pairs (n = 54)* |  |  |  |
| Primary MDS | 32 | WES | 25 |
| MDS/MPN | 14 | WES | 12 |
| AML-MRC | 8 | WES | 6 |
|  |  |  |  |
| *Tumor Only (n = 23)* |  |  |  |
| Primary MDS | 14 | TSCA | Not Done |
| MDS/MPN | 9 | TSCA | Not Done |
| AML-MRC | 0 | TSCA | Not Done |
| Total | 77 |  | 43 |

TSCA, TruSeq Custom Amplicon; WES, Whole-exome sequencing

MDS/AML[31], and adult MDS/AML[32] (Supplementary Datas #8 and 9) and a subset of these variants were classified according to the American College of Medical Genetics (ACMG) criteria[33]. In addition, we reported all loss of function variants (frameshift, nonsense and splice site alterations) in all genes (Supplementary Data #10). In this study we used a VAF cut off of > 40% in the lymphocytes to classify variants as germline, unless there was evidence of CN-LOH in the lymphocytes. We observed putative germline variants in *PTPN11*, *NF1*, and *NRAS* in patients with JMML, which are known associations[30]. Eight patients had germline variants in *SAMD9* or *SAMD9L* (discussed below). Other potentially causative germline variants identified were in *RRAS* and *BRCA2* (Table 2 and Supplementary Data #9). The *RRAS* p.Q87L variant, which has been previously shown to increase MAPK activity and inhibit apoptosis[34,35], has been reported in rapidly progressive JMML, and other germline variants in *RRAS* have been identified in Noonan Syndrome and JMML[36]. Only one patient had a presumed *GATA2* germline variant (p.L375F). Resequencing of intron 4 of *GATA2*, a previously described mutation hotspot in pediatric MDS[15], in all cases of primary MDS subjected to WES did not identify any additional germline events in these 32 cases. Of note, in the 14 primary cases with only tumor material available, there were 2 cases with *GATA2* mutations at a variant allele frequency (VAF) of > 40%, suggesting that they may be germline events (Supplementary Fig. 6). One of these cases had two separate *GATA2* variants at a VAF > 40%. In addition, we identified 2 *PTPN11* variants via targeted amplicon sequencing with VAF's suggestive of germline lesions (Supplementary Fig. 7). Although material for normal comparator gDNA was not available to confirm germline lesions in these patients, the variants with VAF's > 40% were significantly higher than other somatic mutations present in the same patient.

**Germline *SAMD9* and *SAMD9L* variants are frequent in primary MDS.** We previously described a germline *SAMD9* variant (p.E1136Q) in three siblings with isolated familial MDS[23], and others have described *SAMD9* and *SAMD9L* variants as causative lesions in MIRAGE syndrome[19,20] and Ataxia-Pancytopenia Syndrome (APS)[21] or a syndrome resembling APS but with less severe neurological manifestations[22], respectively. In total (including the previously reported three siblings), eight patients (17%) in our primary MDS cohort had presumed *SAMD9* (n = 4) or *SAMD9L* (n = 4) germline variants (RCC: 7, RAEB: 1). Within the primary MDS cohort, 42% of the patients with loss of material on chromosome 7 have germline *SAMD9* or *SAMD9L* variants. With the exception of the *SAMD9* p.E1136Q variant, all variants identified were previously unreported missense variants (*SAMD9*: p.T778I; *SAMD9L*: p.W1180R, p.S626L (n = 2), and p.R1281K) and have variable predicted impacts on protein function (Fig. 4a and Supplementary Data #11). The two patients (SJ018222 and SJ018225) with the *SAMD9L* p.S626L variants are relatives and culture of bone marrow fibroblasts from SJ018222 confirmed the germline status of this variant. Similar to previous studies evaluating *SAMD9* and *SAMD9L* variants, all variants showed gain-of-function activity that leads to decreased cell proliferation (Fig. 4b, c, and Supplementary Fig. 8). Furthermore, *SAMD9/SAMD9L* gain-of-function variants inhibit the induction of ERK phosphorylation in response to serum (Supplementary Fig. 9). Depending on the extent of monosomy 7 observed in the tumor cells, the VAF of the *SAMD9/SAMD9L* variant decreased accordingly, suggesting a preferential loss of the allele harboring the deleterious variant (Fig. 4d). Previous studies showed the existence of acquired mutations in *SAMD9* or *SAMD9L* that can serve to rescue the deleterious effects of a gain-of-function variant. Further, we previously reported on similar mechanisms in the included family with the germline p.E1136Q *SAMD9* variant[23]. In the remaining five patients, we observed a subclonal CN-LOH event in the lymphocytes of SJ018225 that removes the pathologic p.S626L variant (see Supplementary Fig. 5). No other clear rescue mutations were observed in the remaining four patients.

**Ras-MAPK pathway mutations are most common in pediatric MDS.** Genes involved in the Ras/MAPK pathway were the most common mutations (including both germline and/or somatic) in

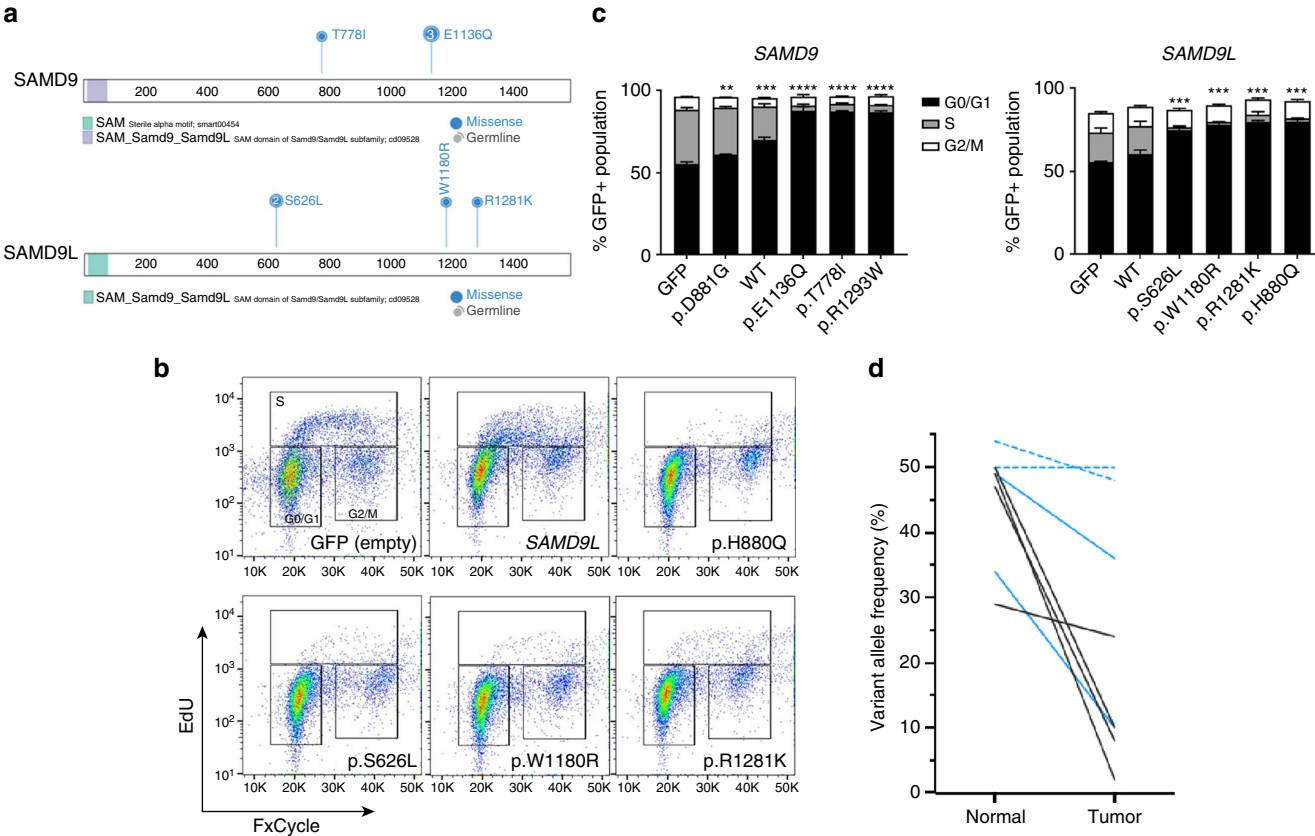

**Fig. 4** Gain-of-function mutations in *SAMD9* and *SAMD9L* decrease cell proliferation and inhibit cycle progression. **a** Schematic showing the protein structure of *SAMD9* and *SAMD9L*. **b** Flow cytometry plots (FxCycle-total DNA content vs. EdU incorporation) showing the cell cycle inhibiting effects of gain-of-function *SAMD9L* mutations. *SAMD9L* p.H880Q (positive control) is a gain-of-function mutation previously reported in Ataxia-Pancytopenia syndrome[21]. **c** EdU incorporation assay showing that gain-of-function *SAMD9/SAMD9L* mutations inhibit cells from progressing through the cell cycle as depicted by the relative absence of cells in S-phase. *SAMD9* p.R1293W (positive control) is a previously reported gain-of-function mutation in MIRAGE syndrome[19]. *SAMD9* p.D881G is a common SNP that is not predicted to be pathogenic. $^{**}$: $p < 0.01$, $^{***}$: $p < 0.001$, $^{****}$: $p < 0.0001$ (student's *t*-test). Error bars indicate standard deviation. Data in B & C are representative of 2–3 experiments completed in triplicate. **d** VAF plot, obtained from targeted deep sequencing, showing the preferential loss (decrease in VAF) of the *SAMD9/SAMD9L* mutations in the tumor population. Black lines: *SAMD9*; Blue lines: *SAMD9L*; dashed lines indicate cases where a subclonal (3/20 metaphases) del(7) was detected only by conventional karyotyping

our cohort, most notably *PTPN11* (15 mutations in 14 patients) and *NRAS* (10 mutations in 9 patients) (Fig. 5a, Supplementary Datas #3, 6, and 9). In addition to the genes commonly mutated in myeloid malignancies (e.g., *NRAS, KRAS, NF1, CBL*, and *PTPN11*), we also identified mutations in *RRAS* (germline), *BRAF*, and *SOS1* (Supplementary Fig. 10). In total, we identified Ras/MAPK mutations in 55% of the total cohort and 43% of primary MDS cases with VAF's ranging from 2 to 85% (Supplementary Fig. 11). Interestingly, Ras/MAPK mutations were enriched in the higher-grade primary MDS, RAEB (RAEB: 65% vs RCC: 17%, $p = 0.002$, Fisher's exact test). In particular for *PTPN11*, the variant was frequently detected in the lymphocyte "normal" sample, suggesting that these are a result of tumor-in-normal contamination, or variants that are either germline or mosaic (Supplementary Fig. 12). Due to low number of somatic coding mutations in these cases it is difficult to confidently make this distinction. This pattern was not limited to JMML cases. The *BRAF* mutations found in our MDS cohort (p.G469A and p.D594N) induce some level of constitutive ERK phosphorylation and confer IL3 independence in the murine Ba/F3 cell line[37] (Fig. 5b, c, and Supplementary Fig. 13).

**Additional somatic mutations in pediatric MDS.** In addition to mutations in the Ras/MAPK pathway, we also observed somatic mutations in *SETBP1*. As previously reported, these mutations were present in both the JMML cohort[30] and MDS cohort[38]. *CEBPA* ($n = 1$, 1%), *RUNX1* ($n = 2$, 3%), and *ETV6* ($n = 4$, 5%) somatic mutations were also identified. Somatic *TP53* mutations, all within the DNA-binding domain, were present in three cases (4%). *DNMT3A, ASXL1*, and *TET2* mutations were notably absent from our cohort. In contrast to adult MDS[5], RNA splicing genes (*SRSF2*: $n = 1$, *U2AF2*: $n = 1$, *U2AF1*: $n = 1$) were rarely mutated in our cohort ($n = 3$, 4%) (Fig. 2c and Supplementary Fig. 14). Collectively, our comprehensive somatic and germline sequencing illustrates new patterns of mutations in pediatric primary MDS (Fig. 6) and highlights differences between MDS/MPN and AML-MRC (Supplementary Figs. 15 and 16). Within the primary MDS cohort, the presence of Ras/MAPK mutations, monosomy 7 or germline *SAMD9/SAMD9L* variants do not impact patient outcome (Supplementary Fig. 17). Consistent with other studies, the presence of *SETBP1* mutations did appear to be associated with inferior outcome despite small numbers of patients[39] (Supplementary Fig. 17f).

**Fusion events are rare in pediatric MDS.** RNA-sequencing of high-quality RNA material from 43 cases demonstrated that fusion events are rare in pediatric MDS. Only 2 fusions (*RUNX1-MECOM* and *CSNK1A1-LECT2*) were identified in 25 primary MDS patients (Supplementary Fig. 18). *RUNX1-MECOM* is a known fusion in AML/MDS[40]. The *CSNK1A1-LECT2* fusion is

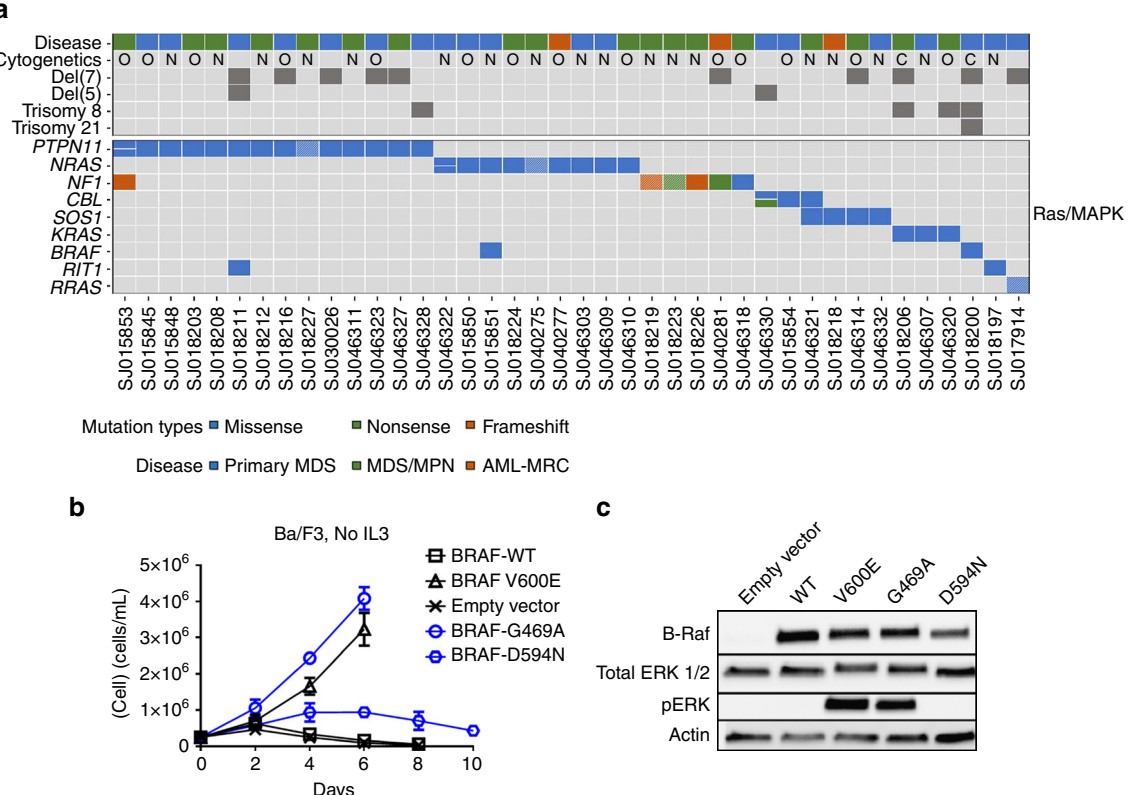

**Fig. 5** Ras/MAPK pathway mutations in pediatric MDS, MDS/MPN, and AML-MRC. **a** Heat map showing all Ras/MAPK pathway mutations, both somatic and presumed germline (cells with hatched lines indicate presumed germline variants from WES tumor/normal cases), in the pediatric MDS cohort ($n = 77$). **b** Growth curves of Ba/F3 cells transduced with retrovirus containing *BRAF* mutations. Blue curves indicate mutations found in the pediatric MDS cohort, and black curves are positive (V600E) and negative controls. Error bars indicate standard deviation. **c** Western blots of BRAF, total ERK, and phosphorylated ERK from lysates of 293 T cells transiently transfected with each *BRAF* mutation. Data are representative of three biological replicates

| Table 2 Presumed germline mutations linked to disease | | | | | | | | |
| --- | --- | --- | --- | --- | --- | --- | --- | --- |
| Case | Diagnosis | Gene | RefSeq Accession | Mutation Type | Nucleotide Change | Amino Acid Change | VAF (lymphocytes) | ACMG Classification |
| SJ018213 | Primary MDS | *BRCA2* | NM_000059 | nonsense | c.G3922T | p.E1308X | 0.47 | P |
| SJ017914 | Primary MDS | *RRAS* | NM_006270 | missense | c.A260T | p.Q87L | 0.47 | VUS |
| SJ015855 | Primary MDS | *SAMD9* | NM_017654 | missense | c.G3406C | p.E1136Q | 0.47 | VUS |
| SJ015856 | Primary MDS | *SAMD9* | NM_017654 | missense | c.G3406C | p.E1136Q | 0.5 | VUS |
| SJ018211 | Primary MDS | *SAMD9* | NM_017654 | missense | c.C2333T | p.T778I | 0.49 | VUS |
| SJ018228 | Primary MDS | *SAMD9* | NM_017654 | missense | c.G3406C | p.E1136Q | 0.29 | VUS |
| SJ018198 | Primary MDS | *SAMD9L* | NM_152703 | missense | c.T3538C | p.W1180R | 0.5 | VUS |
| SJ018222 | Primary MDS | *SAMD9L* | NM_152703 | missense | c.C1877T | p.S626L | 0.54 | VUS |
| SJ018225 | Primary MDS | *SAMD9L* | NM_152703 | missense | c.C1877T | p.S626L | 0.34 | VUS |
| SJ040280 | Primary MDS | *SAMD9L* | NM_152703 | missense | c.G3842A | p.R1281K | 0.49 | VUS |
| SJ018219 | MDS/MPN | *NF1* | NM_000267 | frameshift | c.3457_3460del | p.L1153_N1154fs | 0.51 | P |
| SJ018223 | MDS/MPN | *NF1* | NM_000267 | nonsense | c.C2446T | p.R816X | 0.51 | P |
| SJ040275 | MDS/MPN | *NRAS* | NM_002524 | missense | c.G34A | p.G12S | 0.44 | LP |
| SJ018227 | MDS/MPN | *PTPN11* | NM_002834 | missense | c.A182G | p.D61G | 0.47 | P |
| SJ018203 | MDS/MPN | *RUNX1* | NM_001754 | missense | c.C425A | p.A142D | 0.51 | VUS |
| SJ040268 | AML-MRC | *GATA2* | NM_001145661 | missense | c.C1123T | p.L375F | 0.49 | P |

VAF, variant allele frequency; P pathogenic; LP, likely pathogenic; VUS, variant of unknown significance
Includes only mutations identified by WES in 54 tumor/normal pairs. Note: the VAFs for SJ018228 and SJ018225 are <40% because a LOH event is present

the result of an intrachromosomal deletion of chromosome 5 and results in a fusion transcript that removes the kinase domain of *CSNK1A1*, and is likely similar to other previously reported loss of function alterations in *CSNK1A1*[41]. Two presumed novel fusion transcripts (*NUP98-JADE2* and *SNRNP70-FGFR1*) were identified in patients with MDS/MPN. JADE2 contains two Plant

Homeo-Domains (PHD), which are common features for translocation partners with *NUP98*[42], and thus is predicated to function similar to *NUP98*-PHD fusions, like *NUP98-NSD1* and *NUP98-KDM5A*[43]. Likewise, the *SNRNP70-FGFR1* preserves the kinase domain of *FGFR1*, like other FGFR1 translocations in cancer[44]. Four fusion events were detected and validated for

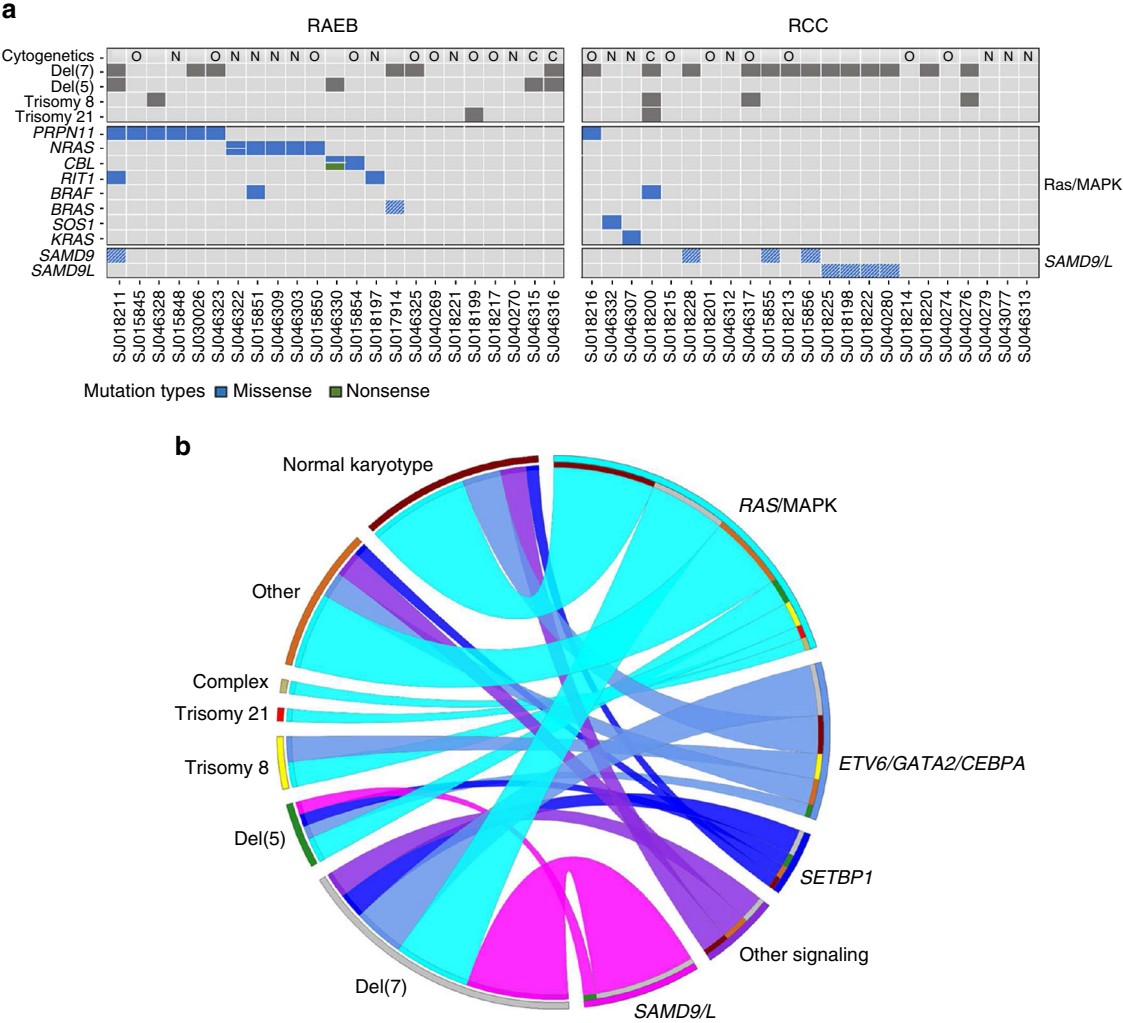

**Fig. 6** The genomic landscape of pediatric primary MDS. **a** Heat map indicating primary MDS patients, subdivided into RCC and RAEB categories, with somatic mutations, germline variants (cells with hatched lines), and transcript fusions. Ras/MAPK mutations are enriched in the RAEB subgroup (65% vs 17%, $p = 0.002$, Fisher's exact test) **b** Ribbon plot showing associations between cytogenetic abnormalities and recurrent mutations in myeloid neoplasms. Data from WES and targeted amplicon sequencing of the primary MDS cohort ($n = 46$) was used to build the plot. Associations between a cytogenetic abnormality and a mutation are connected by a ribbon, the width of which is proportional to the frequency of that association

AML-MRC. With the exception of an *NF1-RHOT1* fusion that would result in *NF1* loss-of-function, the remaining 3 fusions identified (*NUP98-KDM5A*, *DEK-NUP214*, *ETV6-MECOM*) are known to be associated with myeloid malignancies (Supplementary Fig. 19).

## Discussion

Previous studies on pediatric MDS have focused mostly on gene panels[31,38], which have been guided based on our understanding of adult MDS and inherited bone marrow failure syndromes. In this study we sought to more comprehensively define genetic variation in pediatric MDS through a combination of WES, amplicon sequencing, and RNA-sequencing and our data show a complex pattern of germline and somatic mutations (Fig. 6). We also expanded our analyses to include other pediatric myeloid neoplasms with potentially overlapping morphologic (e.g., dysplasia) or cytogenetic (e.g., del (7)/(7q)) features, such as juvenile myelomonocytic leukemia ($n = 19$), unclassifiable myelodysplastic/myeloproliferative neoplasms ($n = 4$), and acute myeloid leukemia with myelodysplasia-related changes ($n = 8$).

We identified germline variants in *SAMD9* or *SAMD9L* in 8 of the 46 cases of primary MDS. *SAMD9* and *SAMD9L* are sterile

acid domain containing genes located at 7q21[45], in a region frequently deleted in myeloid tumors[46,47], and their protein products have been implicated in endosomal function and interferon signaling[22,48]. Loss of murine *Samd9l* can lead to a MDS-like disease in mice[49]. Recently, germline variants in these genes have been associated with multisystem disorders with a range of systemic abnormalities, including neurologic, endocrine, gastrointestinal, immune, and hematopoietic systems[19,21,22]. A subset of these patients also developed MDS with monosomy 7. It has been proposed that loss of chromosome 7 is a cellular adaptation to the growth suppressive properties of the mutant SAMD9 or SAMD9L protein-referred to previously as "adaptation by aneuploidy"[19]. In all cases, we similarly see selective loss of the chromosome that harbors the heterozygous germline variant. The patients included in this study lack the profound extra-hematopoietic phenotypes of MIRAGE and Ataxia-Pancytopenia patients, rather they appear to have isolated MDS. Seven of the 8 patients presented with RCC and had few somatic cooperating events (total somatic mutations: 26, range: 0–10/patient), including 2 cases with no somatic coding mutations. The one patient with RAEB had multiple somatic events, including mutations in *PTPN11*, *ETV6*, *SETBP1*, and deletion of

chromosome 5 (resulting in the previously mentioned *CSNK1A1-LECT2* fusion). Although our findings here clearly need to be expanded to a larger cohort, we suggest that germline variants in *SAMD9* and *SAMD9L* are a new class of lesions that need to be screened for in children with MDS and monosomy 7.

In total from the cases with tumor and normal material, we identified 10 primary MDS cases (out of 32) with germline variants that may be causative (Table 2). Interestingly, none of these variants were in *GATA2*, although in 14 primary MDS cases with tumor-only material available there were 3 additional *GATA2* variants (1 case with 2 separate mutations) with VAF's > 40% suggesting that they may be germline events. One AML-MRC patient had a germline *GATA2* variant. Five MDS/MPN patients, all JMML, had germline variants, 4 of which were in genes of the Ras/MAPK pathway. Surprisingly, we did not find any germline variants in genes implicated in bone marrow failure syndromes that have overlap with MDS, such as dyskeratosis congenita[12,50]. Of note, our collective findings do show differences in the genomic profiles of pediatric MDS from our single institution cohort and those published by other groups. In addition to fewer germline *GATA2* variants, we also observed fewer somatic *SETBP1* mutations. We suggest these differences are largely due to the heterogeneous nature of MDS coupled with small cohort sizes.

Monosomy 7 is a known cytogenetic abnormality in many myeloid tumors, and previous studies have demonstrated that it occurs in nearly 25% of children with MDS[15,51]. The presence of monosomy 7 has also been associated with somatic mutations in *SETBP1*[38] and germline alterations in *GATA2*[15]. Our analyses confirmed the high frequency of deletions involving all or portions of chromosome 7. In addition, we also observed an association between deletions involving chromosome 7 and two distinct groups (Fig. 6b). In particular, 100% of patients with *SAMD9/SAMD9L* mutations (7/8 were classified as RCC), and 71% of RAEB patients with a Ras/MAPK mutation were associated with chromosome 7 deletions. Clearly, larger studies will be needed to confirm these associations.

We further demonstrated that mutations in the Ras/MAPK pathway are more common in pediatric MDS than adult MDS, especially in children with RAEB. Makishima et al. recently reported WES data for 124 adult MDS patients and Ras/MAPK mutations were present in only 10% of those cases[7] vs. our 45% of pediatric primary MDS. Additionally, the mean mutation frequency per patient (pediatric: 5/case vs. adult: 11.4/case) and the frequency of deletions involving chromosome 5 (pediatric: 5% vs. adult: 18%) were different. These differences are not surprising given the disparate morphologic and clinical characteristics of pediatric and adult MDS. Our data are consistent with the recent study by Pastor et al., but due to the more comprehensive sequencing in our study, we identified mutations in other Ras/MAPK pathway genes that may not be included in typical panels for genomic testing[38]. Germline and somatic Ras/MAPK mutations largely define JMML[24,30,52,53] and our data suggest a higher degree of genomic similarity between JMML and pediatric RAEB than previously appreciated, which could have beneficial clinical implications given the new clinical trial with a MEK inhibitor in JMML (Children's Oncology Group ADVL1521). Pediatric MDS is not only contrasted with adult MDS or chronic myelomonocytic leukemia (CMML) in regard to Ras/MAPK pathway mutations but also in mutations of genes encoding epigenetic regulators (more frequent in adult MDS/CMML), thus potentially suggesting that the epigenetic landscape of pediatric MDS is more permissive to transformation than adults.

In summary, we provide the first comprehensive view on the genomic landscape of pediatric MDS. In addition to increasing the spectrum of somatic mutations in pediatric MDS, some of which suggest genomic similarity to JMML, we have expanded the list of genes with potential germline variation in children with MDS. In particular, we define *SAMD9* and *SAMD9L* as new genes linked to childhood MDS.

## Methods

**Patient sample details**. Tumor and germline samples, when applicable, were obtained with informed consent using a protocol approved by the St. Jude Children's Research Hospital Institutional Review Board. All patients with a diagnosis of MDS, MDS/MPN, and AML-MRC between 1988 and 2016 were originally evaluated for sample adequacy. Diagnoses were reviewed by a hematopathologist (J.M.K.) and classified according to WHO 2008 criteria[54]. Detailed clinicopathological information is available in Supplementary Data #1. Samples were de-identified before nucleic acid extraction and analysis. The study cohort comprises 46 primary MDS (23 refractory cytopenia of childhood/RCC and 23 refractory anemia with excess blasts/RAEB), 23 MDS/MPN (including 19 with juvenile myelomonocytic leukemia/JMML), and 8 AML-MRC cases for a total cohort of 77 patients. Germline samples were obtained from flow sorted total lymphocytes or CD3 + T-cells from the diagnostic bone marrow samples (Supplementary Fig. 2). Cryopreserved bulk bone marrow cells were thawed in a 37 °C water bath and transferred to 20% FBS in PBS to remove residual DMSO according to standard approaches[55]. Cells were lysed with ACK lysing buffer (ThermoFisher A1049201) and washed with PBS prior to staining. The following antibodies were used to immunophenotype the cells and facilitate flow sorting of myeloid and lymphoid populations: CD15-FITC (eBioscience, clone HI98), CD71-BV711 (BD Biosciences, clone M-A712), CD34 PE (Beckman, clones QBEnd10, Immu133, Immu409), CD45R-PerCP-Cy5.5 (eBioscience, clone RA3-6B2), CD235a-PE-Cy7 (BD Biosciences, clone GA-R2), CD3-APC-Cy7 (BD Biosciences, clone SK7), CD33-APC (eBioscience, clone WM-53).

**Whole-exome and RNA-sequencing and analysis**. DNA and RNA material was isolated from bulk myeloid or isolated lymphocytes by standard phenol:chloroform extraction and ethanol precipitation. Whole-exome sequencing was completed using the Nextera Rapid Capture Expanded Exome reagent (Illumina) and analyzed as previously described[56]. RNA-sequencing was performed using TruSeq Stranded Total RNA library kit (Illumina) and analyzed, as previously described[56]. Structural variation detection was carried out using CICERO[57], a novel algorithm that uses de novo assembly to identify structural variation in RNA-seq data and Chimerascan[58]. All identified fusions were validated by RT-PCR. Mapping statistics and coverage data are described in Supplementary Data #2. Recurrent SNVs identified via WES were validated by custom amplicon sequencing using the MiSeq platform as previously described[56] (Supplementary Data #12). All SNVs identified by WES and subsequently validated are summarized in Supplementary Data #3. For the 23 cases without a matched normal sample, DNA from whole bone marrow was analyzed using a TruSeq Custom Amplicon (Illumina) approach (Supplementary Data #4 for gene/target list). VarScan 2[59] was used for variant calling on the TruSeq Custom Amplicon data with the following criteria: MAPQ > = 1; minimum read depth at a position to make a call > = 100; minimum supporting reads at a position to call variants > = 10; minimum base quality at a position to count a read > = 23; VAF > = 0.02. The calls with reads showing strong bias or present in majority of the samples were filtered out, and the remaining ones were manually reviewed. The targeted sites (Supplementary Data #4) were also scanned regardless of the above cutoffs for manual review. Further, the coding regions for *SAMD9* and *SAMD9L* were sequenced using a modified 16 S library protocol. Short amplicons were used to validate the known variants. Oligonucleotides were designed to amplify an ~350 bp fragment surrounding each variant and these oligonucleotides included the following Illumina adapters: TCGTCGGCAGCGT-CAGATGTGTATAAGAGACAG-[forward primer] and GTCTCGTGGGCTCGG AGATGTGTATAAGAGACAG-[reverse primer]. The amplicons were then purified with Ampure XP beads, PCR-amplified (five cycles) to attach indices and adapters, followed by an additional purification with Ampure XP beads and quality assessment on a LabChip GX. The samples were run on a MiSeq with a 500cycle nano kit. Discovery sequencing to identify any coding *SAMD9* or *SAMD9L* mutations in cases not subjected to WES was performed by amplifying ~1.5 kb regions with > 50 overlapping bases. The amplicons were submitted to the Hartwell Center for Nextera XT library preparation, per the manufacturer's protocol, and sequenced on a MiSeq with a 500cycle nano kit.

**CNA detection using whole-exome sequencing data**. Samtools[60] mpileup command was used to generate an mpileup file from matched normal and tumor BAM files with duplicates removed. VarScan2 was then used to take the mpileup file to call somatic CNAs after adjusting for normal/tumor sample read coverage depth and GC content. Circular Binary Segmentation algorithm[61] implemented in the DNAcopy R package was used to identify the candidate CNAs for each sample. B-allele frequency info for all high quality dbSNPs heterozygous in the germline sample was also used to assess allele imbalance.

**Germline analysis**. Whole exome sequencing (WES) data were analyzed using internal workflow that were previously described[29]. Briefly, the sequencing data were analyzed for the presence of single-nucleotide variants and small insertions and deletions and for evidence of germline mosaicism. Germline copy-number variations and structural variations were identified with the use of the Copy Number Segmentation by Regression Tree in Next Generation Sequencing (CONSERTING)[62] and Clipping Reveals Structure (CREST)[63] algorithms. For all SNPs and INDELs, functional prediction (e.g., SIFT, CADD, and Polyphen) and population minor allele frequency (MAF) were annotated. In this work, 3 databases were used for population MAF annotation: (i) NHLBI GO Exome Sequencing Project (http://evs.gs.washington.edu/EVS/); (ii) 1000 genomes (http://www.internationalgenome.org); and (iii) ExAC non-TCGA version (http://exac.broadinstitute.org/). A gene list of 1176 genes were composed from various resources: (i) literature review of genes that are potentially involved in AML, MDS, inherited bone marrow failure syndromes, as well as other cancer types[16,29,31,32,64] (ii) genes that were involved in splicing from predefined pathways (e.g., splicing) in KEGG, GeneOntology, Reactome, Gene Set Enrichment Analysis (GSEA), and NCBI (Supplementary Data #6). Novel or extremely rare variants (MAF ≤ 0.1%) that passed sequence quality check were kept for subsequent analysis. All non-synonymous mutations in 207 of these genes were comprehensively reviewed and classified according to ACMG guidelines[33] by a medical geneticist (C.K.). Non-synonymous mutations in the remaining 969 genes were merely reported (Supplementary Data #9). In addition, nonsense, frameshift, or splicing mutations in all other genes covered by the exome capture were also reported (Supplementary Data #10). Relevant germline variants were validated by targeted resequencing.

**Cloning and mutagenesis**. pBABE-bleo[BRAFV600E] was a gift from Christopher Counter via Addgene (#53156). BRAF p.V600E was reverted back to wild-type (NM_004333) using the GENEART® Site Directed Mutagenesis System (Invitrogen). The patient specific mutations found in our MDS cohort were then introduced into the wild-type BRAF containing vector. The pCMV6-Entry[SAMD9] (RC219076) and pCMV6-Entry[SAMD9L] (RC207886) vectors were purchased from Origene. SAMD9 and SAMD9L were PCR amplified such that the resultant amplicons contained a 5′-CACC overhang to allow for direct cloning into the Gateway entry vector, pENTR/D-TOPO. SAMD9/L GFP fusions were created by using the Gateway LR clonase II reaction with the pcDNA6.2/N-EmGFP-DEST Gateway destination vector which was purchased from Lifetech (V35620). The pcDNA6.2/N-EmGFP/GW/CAT vector was used as the GFP-empty vector control. Mutagenesis and PCR amplification primers mentioned above are listed in Supplementary Data #13.

**Cell culture, transient transfection, & stable transduction**. For BRAF functional assays murine-specific retrovirus containing BRAF constructs was produced via transient transfection (FuGene®, Promega) of HEK-293T cells (ATCC) with the construct of interest and the EcoPak viral packaging vector, according to standard procedures[65]. Briefly, 48 h following transfection Ba/F3 cells were transduced with viral particles via spinfection/polybrene. For spinfection, 1.5 mL of virus particle containing supernatant from 293 T culture was mixed with ~2 × 10[6] Ba/F3 cells in a total volume of 3 mL of media containing HEPES buffer and 2uL of polybrene. Cells and virus were then centrifuged at 2000×g at 30 °C for 90 min, after which they were allowed to rest at 37 °C, 5% CO$_2$ for an additional 90 min. Following the rest period, the cells were washed with PBS, centrifuged, and suspended in fresh media (RPMI 1640, 10% FBS). After 48 h of transduction Ba/F3 cells were counted via hemacytometer and trypan blue exclusion every 2 days. For SAMD9/SAMD9L functional assays 293 T cells were transiently transfected with constructs of interest (FuGene®, Promega).

**Dye dilution cell proliferation assays**. Prior to SAMD9/SAMD9L transfection 293 T cells were stained with either CellTrace™ Violet (for dye dilution experiment only) or CellTrace™ Yellow (for combination dye dilution and EdU incorporation cell cycle assays) (Invitrogen) according to the manufacturer's protocol. In total, 72 h following transfection 293 T cells were collected, fixed with 4% paraformaldehyde, and analyzed by flow cytometry.

**EdU cell cycle assays**. Following a 24–48 h transfection with SAMD9/L 293 T cells were treated with 10uM EdU for 2 h. Following the EdU incubation, 293 T cells were harvested and fixed with 4% paraformaldehyde. After fixing, the Click-It® reaction (Invitrogen, C10635) was performed according to the manufacturer's protocol. Following the Click-It® reaction, total DNA was labeled with FxCycle (Invitrogen, F10347), and analyzed by flow cytometry.

**Immunoblotting and serum stimulation**. For BRAF studies, 48hrs following viral transduction, Ba/F3 cell lysates were prepared with Laemmli buffer and separated on standard polyacrylamide gels. For SAMD9/SAMD9L studies, the induction of phosphorylated ERK was assayed following serum stimulation. Twenty-four h after SAMD9/L transfection, 293 T media (10% FBS) was replaced with serum deficient (1% FBS) media for 18–24 h. Subsequently, cell lysates were prepared with Laemmli buffer at three time points: 0 min (prior to 10% FBS replacement) and 10 and 60 min following full media replacement. The following antibodies were used

at a 1:1000 dilution for immunoblotting: SAMD9 (abcam, ab180575), GFP (Invitrogen, A11122), BRAF (Santa Cruz Biotechnology, sc-5284), and from Cell Signaling Technologies: Total ERK (4695S), Phos-ERK (9101S), and GAPDH (2118S).

**Statistical methods**. The student's t-test, two-tailed, assuming equal variances, was used when comparing two experimental groups (e.g., SAMD9/SAMD9L mutations) or diagnostic subgroups (e.g., mutation frequency). The Fisher's exact test was used to compare the frequency of Ras/MAPK mutations between RAEB and RCC subgroups. Overall survival was defined as the time difference between the date of MDS, MDS/MPN, or AML-MRC diagnosis and the date of death. Patients who were alive at the time of last follow up were considered censored. Survival curves between groups were compared via log-rank tests.

**Data availability and accession codes**. Genomic data have been deposited in the European Genome-phenome Archive (EGA), which is hosted by the European Bioinformatics Institute (EBI), under accession EGAS00001002202. All other remaining data are available within the Article and Supplementary Files, or available from the authors upon request.

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

## Acknowledgements

We thank all the patients and their parents at SJCRH for contributing material for this study. We thank the Biorepository, the Flow Cytometry and Cell Sorting Core, and the Hartwell Center for Bioinformatics and Biotechnology of St. Jude Children's Research Hospital. Jinghui Zhang and the Department of Computational Biology graciously provided previously established bioinformatic pipelines and support. We thank Drs. Timothy Ley and Matthew Walter for critical review of this manuscript. This work was funded by the American Lebanese and Syrian Associated Charities of St. Jude Children's Research Hospital and grants from the US National Institutes of Health (P30 CA021765, Cancer Center Support Grant, and K08 HL116605 (J.M.K.)). J.M.K. holds a Career Award for Medical Scientists from the Burroughs Wellcome Fund.

## Author contributions

J.R.S., J.M., T.L., and J.M.K. prepared the manuscript. J.R.S., T.L., V.B., and J.M.K. were responsible for experimental design and analysis. T.L. prepared DNA and RNA from all patient samples. J.M., M.W., S.W., G.S., G.W., J.E. were responsible for bioinformatic data analysis. C.K., K.E.N., and J.M.K. analyzed germline variants and determined their likely pathogenicity. C.G.M. and R.C.R. assisted with data analysis and acquisition of patient cases.

## Additional information

**Competing interests:** The authors declare no competing financial interests.

