## [Peer Review File · Nature Communications]

Reviewers' comments:

Reviewer #1 Expert in leukemogenesis:

This submission by Schwartz and colleagues describes comprehensive genome-wide sequencing, copy number, and cytogenetic analysis on a substantial cohort of patients accrued at St Jude over many years. A great strength of this work is that it is unbiased and thorough. However, the somewhat atypical nature of the population of patients referred to St. Jude is a minor concern and the authors appropriately emphasize the need for further validation studies. This is nevertheless an important contribution to the field. I have the following comments for the author's consideration.

(1) The use of sorted CD3+ lymphocytes as a source of "normal" germline DNA is a reasonable approach in the absence of skin (or bone marrow) fibroblasts or similar comparative tissues. However, it is imperfect, and the authors acknowledge this. While there is clearly no issue when the MDS bone marrow shows a mutation that is absent in CD3+ T cells, this becomes problematic for distinguishing true germline mutations from a somatic mutation arising in the stem/progenitor compartment that affects both myeloid and lymphoid cells. There are a couple of examples of this in the older JMML literature that could be cited (e.g. Blood 1996, 88, 4314 and Blood 2000, 96, 2310). With respect to putative "germline" mutations listed in Table 2, I have some concerns about three of them. The WT1 mutation is present at a fairly low allele frequency and I am skeptical that this is a germline mutation unless the patient had phenotypic evidence of WAGR syndrome or developed Wilms' tumor. The E76K PTPN11 mutation is very strongly activating, was not reported as a Noonan mutation in the paper of Kratz, Loh and their colleagues (Blood 2005, 106, 2183), and is also present at a subclonal allele frequency (0.39). The G12S NRAS is perhaps the most difficult since I recall that patients with this mutation may have "spontaneous" resolution of JMML and it is present at a fairly high allele frequency. Has this mutation been reported on Noonan's or in any of the other Rasopathy disorders? Did this patient have any developmental phenotypes? My intention here is to not disparage the approach of using CD3+ T cells as a source of DNA for comparison studies, which certainly adds great value. I only raise questions about a few of these cases that seem equivocal to encourage the authors to review them and/or comment about one or more of them specifically.

(2) Age at onset is something that is of interest to clinicians and scientists who are interested in pediatric MDS. While these data are included in Table S1, I think it would be useful to calculate and present the age at diagnosis for the JMML, RCC, and RAEB patients in the Results section. It looks that the JMML patients are younger (not unexpected) than those with RCC or RAEB.

(3) In line 106, I think it should be "refractory anemia (not cytopenia).....RAEB"

(4) One comment about the monosomy 7 data, which is generally presented very well. It appears that there might be two distinct associations: (1) with SAMD9/9L mutations in RCC; and, (2) with Ras pathway mutations in RAEB. The authors partially address this in the Results, but it would be useful to come back to this point in the Discussion.

(5) The authors might mention that the high frequency of Ras pathway mutations in non-JMML pediatric MDS has potential clinical implications as therapeutic trials of MEK inhibitors will open soon in JMML. These trials follow-up on interesting work in mouse models of JMML. Children with MDS might also be candidates similar treatment if this is shown to be beneficial in JMML.

(6) SETBP1 mutations appear to track with RAEB in pediatric MDS and there is a suggestion that they predict relapse in JMML patients undergoing stem cell transplants (Blood 2014, 124, 410). Were any of the patients in this series transplanted and, if so, was there any association of outcome with a SETBP1 mutation?

(7) A striking finding of this work and of studies comparing the genomics of JMML versus CMML is the lack of mutations in epigenetic regulators in pediatric patients versus adults with MDS/CMML. The authors might comment on this in the Discussion as the mutational spectrum may offer clues into the important question of "soil" in transformation. Perhaps hematopoietic stem cells are in a more permissive epigenetic state for transformation by Ras early in life than in adulthood?

Reviewer #2 Expert in paediatric haematology:

The manuscript reported whole-exome sequencing/targeted sequencing and/or RNA sequencing of 48 pediatric MDS, including 24 RCC cases and 24 RAEB cases, and related myeloid neoplasms, such as JMML (N=18). This is an intensive study of an unprecedentedly large cohort of pediatric MDS, and would make a significant contribution to this field. The authors revealed the genetic landscape of RCC and RAEB, where pediatric RAEB was characterized by frequent genetic alterations involving Ras/MAPK pathway as with the case of JMML. They also identified frequent (17%) germline mutations in SAMD9/SAMD9L in pediatric MDS patients, and monosomy or copy neutral LOH of chromosome 7q including these genes were also frequently observed in tumor cells. Functional studies showed the gain-of-function nature of SAMD9/SAMD9L variants. However, there are some concerns about the significance of germline variants, and I have a few suggestions that I think that if feasible would improve the work.

Major comments:

- Five germline SAMD9/SAMD9L variants were identified in 8 pediatric MDS patients, of which 4 were novel and classified as "variant of unknown significance" by ACMG criteria. Although authors performed the biological experiments to show functionality of these variants, pathogenicity and significance of these variants should be more carefully described, because the following facts raise a question about their pathogenicity:

- a. Supplementary table 11 showed that most of these novel variants were predicted as "tolerated (T)" or "benign (B)" by well-known prediction tool, such as SIFT (100%, 4/4) and Polyphen2 (75%, 3/4)

- b. SAMD9L S626L variant is separately located from previously reported SAMD9L variants (Tesi et al., Blood) and other variants in this paper

- c. VAF of SAMD9L variants in SJ018222 and SJ040280 were comparable between tumor and germline samples

- d. no family history and related symptoms of MIRAGE syndrome and APS have been identified in patients with these novel mutations

- For samples without germline material, variants with >40% of VAF were referred as germline variants. The authors should show that VAFs of "germline" variant were actually higher than others somatic variants.

- Definition of germline variants is unclear. How authors discriminate germline variants from contamination of tumor cells?

Minor comments:

- The authors described copy number neutral events (N = 3) and frequent PTPN11 mutations in the lymphocyte "normal" sample. They should try to discriminate whether these are derived from contamination of tumor cells or mosaicism. In the presence of contamination of tumor cells, all somatic variants and copy number alterations would be detected in the lymphocytes.

- One RCC patient with compound heterozygous MPL germline variants (p.R102P and p.F104S) was reported in the main text. MPL has been known to causative gene of congenital amegakaryocytic thrombocytopenia, and, therefore, clinical history of this patients should be written in more detail.

-Authors described CN-LOH event or PTPN11 mutation in the lymphocytes "normal" sample. They should try to discriminate whether these are a result of tumor-in normal contamination, or variants that are either germline or mosaic.

-RAEB and JMML showed the genetic similarity such as frequent Ras/MAPK mutations. The authors should analyze the prognostic impact of PTPN11 mutation or secondary mutation such as SETBP1 mutations in RAEB, because these are known poor prognostic factors in JMML. Also, it is interesting to see whether RAEB cases with RAS/MAPK mutations have similar clinical features with JMML, such as higher number of monocytes.

-Some figures and tables seem to be redundant, such as Figure5a and Figure2 as well as Figure1 and Table1.

Thank you for your positive comments regarding our manuscript entitled “*The Genomic Landscape of Pediatric Myelodysplastic Syndromes.*” Please find our point-by-point responses to the reviewers’ comments below.

Although not in response to a specific reviewer we realized that case SJ046327 was miscategorized as RAEB instead of JMML. This change has been made throughout the manuscript.

Reviewer #1 Comments:

This submission by Schwartz and colleagues describes comprehensive genome-wide sequencing, copy number, and cytogenetic analysis on a substantial cohort of patients accrued at St. Jude over many years. A great strength of this work is that it is unbiased and thorough. However, the somewhat atypical nature of the population of patients referred to St. Jude is a minor concern and the authors appropriately emphasize the need for further validation studies. This is nevertheless an important contribution to the field.

We thank this reviewer for their helpful and positive comments regarding our manuscript. Like many genomic studies, we agree on the need for further validation and we anticipate that numerous independent groups are/will be working to validate and extend our findings.

I have the following comments for the author’s consideration.

- 1) The use of sorted CD3+ lymphocytes as a source of “normal” germline DNA is a reasonable approach in the absence of skin (or bone marrow) fibroblasts or similar comparative tissues. However, it is imperfect, and the authors acknowledge this. While there is clearly no issue when the MDS bone marrow shows a mutation that is absent in CD3+ T cells, this becomes problematic for distinguishing true germline mutations from a somatic mutation arising in the stem/progenitor compartment that affects both myeloid and lymphoid cells. There are a couple of examples of this in the older JMML literature that could be cited (e.g. Blood 1996, 88, 4314 and Blood 2000, 96, 2310). With respect to putative “germline” mutations listed in Table 2, I have some concerns about three of them. The *WT1* mutation is present at a fairly low allele frequency, and I am skeptical that this is a germline mutation unless the patient had phenotypic evidence of WAGR syndrome or developed Wilms’ tumor. The E76K *PTPN11* mutation is very strongly activating, was not reported as a Noonan mutation in the paper of Kratz, Loh and their colleagues (Blood 2005, 106, 2183), and is also present at a subclonal allele frequency (0.39). The G12S *NRAS* is perhaps the most difficult since I recall that patients with this mutation may have “spontaneous” resolution of JMML, and it is present at a fairly high allele frequency. Has this mutation been reported on Noonan’s or in any of the other rasopathy disorders? Did this patient have any developmental phenotypes? My intention here is to not disparage the approach of using CD3+ T cells as a source of DNA for comparison studies, which certainly adds great value. I only raise questions about**

a few of these cases that seem equivocal to encourage the authors to review them and/or comment about one or more of them specifically.

We agree with this reviewer regarding their comments on the use of sorted T-cells/lymphocytes as a source of normal genomic DNA. As stated, for most patients only cryopreserved bone marrow cells were available for analysis, which severely limited our options. We appreciate the potential complications with “*distinguishing true germline mutations from a somatic mutation if that somatic mutation is arising from a stem/progenitor compartment that affects both myeloid and lymphoid cells.*” We have added the suggested references regarding the previous reports of somatic mutations found in progenitor cells in JMML (see Lines 149-150).

Regarding specific patients and “germline” mutations found in Table 2, see below.

- a. SJ015851, *WT1* p.E323K: Upon further reading and analysis of our data, we agree that this is likely not a true germline lesion based on the VAFs. As shown in the figure below (blue data point), the VAF of this variant is less than the other “germline” variants in Table 2 and other putative “germline” variants of unknown significance in this patient. Further, we reviewed the charts for this patient, and this patient did not have evidence of Wilm’s tumor or other signs of WAGR syndrome. From a functional standpoint, it is also worth pointing out that most pathologic germline *WT1* mutations result in a truncated protein whereas this patient harbored a missense mutation outside of the Zn-finger domain and thus is less likely to alter protein function. **This variant has been removed from Table 2, and reclassified as somatic.**
- b. SJ018212, *PTPN11* p.E76K: It is difficult to determine if the *PTPN11* mutation in this patient, who was diagnosed with JMML, is truly germline or not. Tumor/normal WES did not identify any high confidence somatic variants for comparison. However, in keeping with our rigid definition of a germline variant requiring at least a VAF of >40% in the lymphocytes, we have reclassified this variant at somatic. **This variant has been removed from Table 2, and reclassified as somatic.**
- c. SJ040275, *NRAS* p.G12S: We agree that this variant is interesting for this patient with JMML. This patient did not have any developmental phenotypes, and the only case reports of this variant are those in patients with spontaneously resolving JMML. While we cannot exclude that this variant merely arose in a hematopoietic stem/progenitor that contributed to lymphoid and myeloid cells, the VAF is greater than 40% (44%) and thus we will leave it as a “germline variant”. Further, the VAF of this variant is similar to other germline polymorphisms in this patient (see below) and is much higher than the lone somatic mutation identified by WES.

- 2) **Age at onset is something that is of interest to clinicians and scientists who are interested in pediatric MDS. While these data are included in Table S1, I think it would be useful to calculate and present the age at diagnosis for the JMML, RCC, and RAEB patients in the Results section. It looks that the JMML patients are younger (not unexpected) than those with RCC or RAEB.**

Supplementary Figure 1a has been moved to the main figures and is now Figure 1b. Median ages have been listed in the figure legend.

- 3) **In line 106, I think it should be “refractory anemia (not cytopenia).....RAEB)”**

We apologize for this oversight. Line 107 has been corrected to read “refractory anemia with excess blast (RAEB).”

- 4) **One comment about the monosomy 7 data, which is generally presented very well. It appears that there might be two distinct associations: (1) with *SAMD9/9L* mutations in RCC; and, (2) with Ras pathway mutations in RAEB. The authors partially address this in the Results, but it would be useful to come back to this point in the Discussion.**

We agree that there does appear to be two main associations with monosomy 7 in this study. We added an additional paragraph focused on monosomy 7 to emphasize this point (see Lines 279-286).

- 5) **The authors might mention that the high frequency of Ras pathway mutations in non-JMML pediatric MDS has potential clinical implications as therapeutic trials of MEK inhibitors will open soon in JMML. These trials follow-up on interesting work in mouse models of JMML. Children with MDS might also be candidates for similar treatment if this is shown to be beneficial in JMML.**

We agree that the high percentage of mutations affecting the Ras pathway in pediatric MDS may have promising therapeutic implications. We added a sentence to stress this point (see lines 297-299).

- 6) ***SETBP1* mutations appear to track with RAEB in pediatric MDS and there is a suggestion that they predict relapse in JMML patients undergoing stem cell transplants (Blood 2014, 124, 410). Were any of the patients in this series transplanted and, if so, was there any association of outcome with a *SETBP1* mutation?**

Our data seem to support findings previously published (Stieglitz et al. *Blood* 2014) as there is a trend toward significance for poorer outcomes if *SETBP1* mutations are present. See Figure S16f. This association could be further confirmed in a larger cohort of pediatric MDS. Please also see the comments below for reviewer 2, who also asked about *SETBP1* mutations and outcomes.

- 7) **A striking finding of this work and of studies comparing the genomics of JMML versus CMML is the lack of mutations in epigenetic regulators in pediatric patients**

versus adults with MDS/CMML. The authors might comment on this in the Discussion as the mutational spectrum may offer clues into the important question of “soil” in transformation. Perhaps hematopoietic stem cells are in a more permissive epigenetic state for transformation by Ras early in life than in adulthood?

We thank the reviewer for this comment. We briefly expanded on this finding. **Please see lines 299-302.**

Reviewer #2 Comments:

The manuscript reported whole-exome sequencing/targeted sequencing and/or RNA sequencing of 48 pediatric MDS, including 24 RCC cases and 24 RAEB cases, and related myeloid neoplasms, such as JMML (N=18). This is an intensive study of an unprecedentedly large cohort of pediatric MDS, and would make a significant contribution to this field. The authors revealed the genetic landscape of RCC and RAEB, where pediatric RAEB was characterized by frequent genetic alterations involving Ras/MAPK pathway as with the case of JMML. They also identified frequent (17%) germline mutations in *SAMD9/SAMD9L* in pediatric MDS patients, and monosomy or copy neutral LOH of chromosome 7q including these genes were also frequently observed in tumor cells. Functional studies showed the gain-of-function nature of *SAMD9/SAMD9L* variants. However, there are some concerns about the significance of germline variants, and I have a few suggestions that I think that if feasible would improve the work.

Major comments:

- Five germline *SAMD9/SAMD9L* variants were identified in 8 pediatric MDS patients, of which 4 were novel and classified as “variant of unknown significance” by ACMG criteria. Although authors performed the biological experiments to show functionality of these variants, pathogenicity and significance of these variants should be more carefully described, because the following facts raise a question about their pathogenicity:

a. Supplementary table 11 showed that most of these novel variants were predicted as “tolerated (T)” or “benign (B)” by well-known prediction tool, such as SIFT (100%, 4/4) and Polyphen2 (75%, 3/4).

Indeed, most of these variants were not predicted to be pathogenic by well-known prediction tools, although very little is known about *SAMD9/SAMD9L* genes and the domain structure of their protein products. A recent article predicted a complex domain architecture of *SAMD9/SAMD9L* through computational dissection¹, but no functional studies. This *in silico* analysis predicted that *SAMD9* and *SAMD9L* likely function as an NTPase, therefore one can imagine particular mutations may effect these processes adversely. Due to this uncertainty we only focused on the *SAMD9* and *SAMD9L* variants not previously reported in ExAc or other databases, and we functionally validated these variants. For comparison, we also functionally tested a presumed non-pathologic common SNP identified in our sequencing studies (*SAMD9* p.D881G and showed that this variant had properties similar to the wild-type proteins, in contrast to the identified variants in this study. Lastly, we also compared our variants to previously reported variants from patients with APS or MIRAGE. We feel strongly that due the unique nature of these germline

variants that functional testing is imperative to determine if *SAMD9/SAMD9L* variants truly alter protein function and can be considered pathologic variants.

b. *SAMD9L* S626L variant is separately located from previously reported *SAMD9L* variants (Tesi et al., Blood) and other variants in this paper.

This variant is indeed in a different region of *SAMD9L* than most of the variants reported by Tesi et al., although the previously reported p.H880Q variant is somewhat close. The p.S626L variant, which was identified in related patients (cousins), both with monosomy 7 and MDS, did clearly alter protein function in our functional studies (Figures 4 & S8). As stated above, the domain structure of *SAMD9L* (and *SAMD9*) are poorly understood. In the article previously mentioned by Mekhedov et al., the 626 position lies between 2 predicted domains (SIR2 and P-loop NTPase).

c. VAF of *SAMD9L* variants in SJ018222 and SJ040280 were comparable between tumor and germline samples.

Monosomy 7 was present in only 3 of 20 metaphases in SJ018222 and SJ040280 and was not detected by WES due to its subclonal nature. The lack of a dramatic decrease in VAF for these lesions in the tumor tissue is expected because of the subclonal nature of monosomy 7.

d. No family history and related symptoms of MIRAGE syndrome and APS have been identified in patients with these novel mutations.

Given that many of our samples were banked decades ago and their associated patients have expired, obtaining family histories of these patients would be very difficult. To our knowledge the patients included in this study did not have associated extra-hematopoietic system symptoms consistent with MIRAGE syndrome or APS.

- For samples without germline material, variants with >40% of VAF were referred as germline variants. The authors should show that VAFs of “germline” variant were actually higher than others somatic variants.

Supplementary Figure 7b has been added and referenced in the results section. This figure shows a significant difference between the VAFs for these “germline” variants (VAF > 40%) and other somatic variants present in those patients.

- Definition of germline variants is unclear. How authors discriminate germline variants from contamination of tumor cells?

As discussed above in the comments for reviewer 1, we appreciate the difficulties with using lymphocytes as a source of “germline” genomic DNA. In this study we used a VAF cut-off of >40% to classify variants as germline, unless there was evidence of CN-LOH in the lymphocytes (as observed for the *SAMD9/SAMD9L* in SJ018225 and SJ018228). In the analysis below, we have included VAF plots for the patients with *PTPN11*, *NRAS*, and *NF1* mutations. Each plot shows all somatic and “germline” calls for that patient, and the blue lines indicate either *PTPN11*, *NRAS*, or *NF1*. The plots that are inside the black boxes are those patients who had an obvious somatic variant without evidence for the variant in the lymphocytes. Notably, the patients without an obvious deleterious “germline” variant (black boxes) tended to have many more mutations as compared to the cases with evidence of the

PTPN11/*NRAS*/*NF1* variants in the lymphocyte fraction. Further, because the patients with potential germline variants in these important genes tended to have less mutation burden overall (average: 5 mutations/patient for total cohort), it is difficult to determine mosaicism with confidence. We made a similar statement in the text (line 205-206). For the stated reason, we opted to leave these data here rather than include as supplemental data, however, are happy to do so if requested.

Minor comments:

- The authors described copy number neutral events (N = 3) and frequent *PTPN11* mutations in the lymphocyte “normal” sample. They should try to discriminate whether these are derived from contamination of tumor cells or mosaicism. In the presence of contamination of tumor cells, all somatic variants and copy number alterations would be detected in the lymphocytes.

See analysis in the comment above.

- One RCC patient with compound heterozygous *MPL* germline variants (p.R102P and p.F104S) was reported in the main text. *MPL* has been known to be the causative gene of congenital amegakaryocytic thrombocytopenia, and, therefore, clinical history of this patient should be written in more detail.

In response to the reviewer’s comments, we re-evaluated the clinical findings in this interesting patient. Although the patient presented at St. Jude with MDS and monosomy 7, we were able to obtain records from the referring hospital which did indeed suggest a history of congenital amegakaryocytic thrombocytopenia (CAMT). This patient’s history from the outside hospital also noted a similarly affected sibling. Although we could not find any documentation of genetic confirmation, we feel it is best to remove this patient from the study as we feel strongly that the study should not include patients with a bone marrow failure syndrome. We apologize for any inconvenience. We have removed all record of this patient from our manuscript.

- Authors described CN-LOH event or *PTPN11* mutation in the lymphocytes “normal” sample. They should try to discriminate whether these are a result of tumor-in normal contamination, or variants that are either germline or mosaic.

Please see analysis above.

- RAEB and JMML showed the genetic similarity such as frequent Ras/MAPK mutations. The authors should analyze the prognostic impact of *PTPN11* mutation or secondary mutation such as *SETBP1* mutations in RAEB, because these are known poor prognostic factors in JMML. Also, it is interesting to see whether RAEB cases with Ras/MAPK mutations have similar clinical features with JMML, such as higher number of monocytes.

1.) As shown in the figures below, *PTPN11* did not have a significant effect on the outcome of RAEB patients, but *SETBP1* did, although these are small numbers (total = 23, and only 3 patients with *SETBP1* mutations) therefore at this point it is difficult to know if there is a true association or not. **Survival curves regarding *SETBP1* have been included in the supplemental materials (Figure S16).**

- 2.) There was not a significant difference of monocyte counts between those RAEB patients with or without Ras/MAPK mutations.

- Some figures and tables seem to be redundant, such as Figure5a and Figure2 as well as Figure1 and Table1.

We apologize that some of the figures seem redundant. This is largely a reflection of our attempts to first show the somatic mutations and then focus on germline variants. Due to the high number of Ras/MAPK mutations, we opted to show both germline and somatic mutations in Figure 5a. Figure 6a has been changed to show only Ras/MAPK and *SAMD9/SAMD9L* mutations which will emphasize the two distinct associations with RAEB and RCC, respectively. Figure 2c does not show the germline variants. Figure 1 does not indicate the sequencing approach for the cohort, therefore we feel that Table 1 contains additional pertinent information.

1. Mekhedov, S.L., Makarova, K.S. & Koonin, E.V. The complex domain architecture of SAMD9 family proteins, predicted STAND-like NTPases, suggests new links to inflammation and apoptosis. *Biol Direct* **12**, 13 (2017).
2. Schwartz, J. *et al.* Germline SAMD9 Mutation in Siblings with Monosomy 7 and Myelodysplastic Syndrome. *Leukemia* **Accepted**(2017).

REVIEWERS' COMMENTS:

Reviewer #1 (Remarks to the Author):

The authors have successfully addressed all of my concerns and I believe that they have also responded to the comments of the other reviewer. This is an excellent study that will likely become a landmark contribution to the field of pediatric MDS and the role of germ line predispositions to these aggressive blood cancers.

Reviewer #2 (Remarks to the Author):

The authors have mostly answered all the questions raised. I have only a few additional comments.

As for my question about the definition of germline variants, the authors said that "In this study we used a VAF cut-off of >40% to classify variants as germline, unless there was evidence of CN-LOH in the lymphocytes (as observed for the SAMD9/SAMD9L in SJ018225 and SJ018228)" in the rebuttal. If they clearly describe this in the main text or method section, it would be helpful for readers to understand the process. In addition, they described that "for the stated reason, we opted to leave these data here rather than include as supplemental data, however, are happy to do so if requested". It would be also useful for readers if they add these figures as supplementary figure.

Reviewer #1 Comments:

The authors have successfully addressed all of my concerns and I believe that they have also responded to the comments of the other reviewer. This is an excellent study that will likely become a landmark contribution to the field of pediatric MDS and the role of germ line predispositions to these aggressive blood cancers.

We thank this reviewer for their positive feedback.

Reviewer #2 Comments:

The authors have mostly answered all the questions raised. I have only a few additional comments.

As for my question about the definition of germline variants, the authors said that "In this study we used a VAF cut-off of >40% to classify variants as germline, unless there was evidence of CN-LOH in the lymphocytes (as observed for the SAMD9/SAMD9L in SJ018225 and SJ018228)" in the rebuttal. If they clearly describe this in the main text or method section, it would be helpful for readers to understand the process. In addition, they described that "for the stated reason, we opted to leave these data here rather than include as supplemental data, however, are happy to do so if requested". It would be also useful for readers if they add these figures as supplementary figure.

We thank this reviewer for their valuable insight and questions that led to further analysis of our data. In the main text we have included a statement of how we defined a presumed germline variant (See Lines 153-155) in the WES cases with tumor and normal data. Furthermore, we have included our somatic and lymphocyte mutation VAF analysis of several patients with *PTPN11*, *NF1*, and *NRAS* mutations in Supplementary Figure 12.